# Stein Variational Gradient Descent as Moment Matching

**Qiang Liu,  Dilin Wang**
Department of Computer Science
The University of Texas at Austin
Austin, TX 78712
{lqiang, dilin}@cs.utexas.edu

## Abstract

Stein variational gradient descent (SVGD) is a non-parametric inference algorithm that evolves a set of particles to fit a given distribution of interest. We analyze the non-asymptotic properties of SVGD, showing that there exists a set of functions, which we call the *Stein matching set*, whose expectations are *exactly* estimated by any set of particles that satisfies the fixed point equation of SVGD. This set is the image of Stein operator applied on the feature maps of the positive definite kernel used in SVGD. Our results provide a theoretical framework for analyzing properties of SVGD with different kernels, shedding insight into optimal kernel choice. In particular, we show that SVGD with linear kernels yields exact estimation of means and variances on Gaussian distributions, while random Fourier features enable probabilistic bounds for distributional approximation. Our results offer a refreshing view of the classical inference problem as fitting Stein's identity or solving the Stein equation, which may motivate more efficient algorithms.

## 1   Introduction

One of the core problems of modern statistics and machine learning is to approximate difficult-to-compute probability distributions. Two fundamental ideas have been extensively studied and used in the literature: variational inference (VI) and Markov chain Monte Carlo (MCMC) sampling (e.g., Koller & Friedman, 2009; Wainwright et al., 2008). MCMC has the advantage of being non-parametric and asymptotically exact, but often suffers from difficulty in convergence, while VI frames the inference into a parametric optimization of the KL divergence and works much faster in practice, but loses the asymptotic consistency. An ongoing theme of research is to combine the advantages of these two methodologies.

Stein variational gradient descent (SVGD) (Liu & Wang, 2016) is a synthesis of MCMC and VI that inherits the non-parametric nature of MCMC while maintaining the optimization perspective of VI. In brief, SVGD for distribution $p(x)$ updates a set of particles $\{x_i\}_{i=1}^n$ parallelly with a velocity field $\phi(\cdot)$ that balances the gradient force and repulsive force,

$$x_i \leftarrow x_i + \epsilon\phi(x_i), \qquad \phi(\cdot) = \frac{1}{n}\sum_{j=1}^{n}\nabla_{x_j}\log p(x_j)k(x_j, \cdot) + \nabla_{x_j}k(x_j, \cdot),$$

where $\epsilon$ is a step size and $k(x, x')$ is a positive definite kernel defined by the user. This update is derived as approximating a kernelized Wasserstein gradient flow of KL divergence (Liu et al., 2017) with connection to Stein's method (Stein, 1972) and optimal transport (Ollivier et al., 2014); see also Anderes & Coram (2002). SVGD has been applied to solve challenging inference problems in various domains; examples include Bayesian inference (Liu & Wang, 2016; Feng et al., 2017),

uncertainty quantification (Zhu & Zabaras, 2018), reinforcement learning (Liu et al., 2017; Haarnoja et al., 2017), learning deep probabilistic models (Wang & Liu, 2016; Pu et al., 2017) and Bayesian meta learning (Feng et al., 2017; Kim et al., 2018).

However, the theoretical properties of SVGD are still largely unexplored. The only exceptions are Liu et al. (2017); Lu et al. (2018), which studied the partial differential equation that governs the evolution of the limit densities of the particles, with which the convergence to the distribution of interest can be established. However, the results in Liu et al. (2017); Lu et al. (2018) are asymptotic in nature and hold only when the number of particles is very large. More recently, Chen et al. (2018) studied non-asymptotic properties of a linear combination of SVGD and Langevin dynamics, whose analysis, however, mainly exploits the propriety of Langevin dynamics and does not work for SVGD alone. A theoretical understanding of SVGD in the finite sample size region is still missing and of great practical importance, especially given that the particle sizes used in practice are often relatively small, thanks to the property that SVGD with a single particle ($n = 1$) exactly reduces to finding the mode (a.k.a. maximum a posteriori (MAP)).

**Our Results**    We analyze the finite sample properties of SVGD. In contrast to the dynamical perspective of Liu et al. (2017), we directly study what properties a set of particles would have if it satisfies the fixed point equation of SVGD, regardless of how we obtain them algorithmically, or whether the fixed point is unique. Our analysis indicates that the fixed point equation of SVGD is essentially a moment matching condition which ensures that the fixed point particles $\{x_i^*\}_{i=1}^n$ *exactly* estimate the expectations of all the functions in a special function set $\mathcal{F}^*$,

$$\frac{1}{n} \sum_{i=1}^{n} f(x_i^*) = \mathbb{E}_p f, \quad \forall f \in \mathcal{F}^*.$$

This set $\mathcal{F}^*$, which we call the *Stein matching set*, consists of functions obtained by applying Stein operator on the linear span of feature maps of the kernel used by SVGD.

This framework allows us to understand properties of different kernels (and the related feature maps) by studying their Stein matching sets $\mathcal{F}^*$, which should ideally either match the test functions that we are actually interested in estimating, or is as large as possible to approximate the overall distribution. This process is difficult in general, but we make two observations in this work:

*i*) We show that, by using linear kernels (features), SVGD can *exactly* estimate the mean and variance of Gaussian distributions when the number of particles is larger than the dimension. Since Gaussian-like distributions appear widely in practice, and the estimates of mean and variance are often of special importance, linear kernels can provide a significant advantage over the typical Gaussian RBF kernels, especially in estimating the variance.

*ii*) Linear features are not sufficient to approximate the whole distributions. We show that, by using random features of strictly positive definite kernels, the fixed points of SVGD approximate the whole distribution with an $O(1/\sqrt{n})$ rate in kernelized Stein discrepancy.

Overall, our framework reveals a novel perspective that reduces the inference problem to either a *regression problem* of fitting Stein identities, or *inverting the Stein operator* which is framed as solving a differential equation called Stein equation. These ideas are significantly different from the traditional MCMC and VI that are currently popular in machine learning literature, and draw novel connections to Quasi Monte Carlo and quadrature methods, among other techniques in applied mathematics. New efficient approximate inference methods may be motivated with our new perspectives.

## 2   Background

We introduce the basic background of the Stein variational method, a framework of approximate inference that integrates ideas from Stein's method, kernel methods, and variational inference. The readers are referred to Liu et al. (2016); Liu & Wang (2016); Liu et al. (2017) and references therein for more details. For notation, all vectors are assumed to be column vectors. The differential operator $\nabla_{\boldsymbol{x}}$ is viewed as a column vector of the same size as $\boldsymbol{x} \in \mathbb{R}^d$. For example, $\nabla_{\boldsymbol{x}} \phi$ is a $\mathbb{R}^d$-valued function when $\phi$ is a scalar-valued function, and $\nabla_{\boldsymbol{x}}^\top \phi(\boldsymbol{x}) = \sum_{i=1}^d \partial_{x_i} \phi(\boldsymbol{x})$ is a scalar-valued function when $\phi$ is $\mathbb{R}^d$-valued.

**Stein's Identity** Stein's identity forms the foundation of our framework. Given a positive differentiable density $p(\boldsymbol{x})$ on $\mathcal{X} \subseteq \mathbb{R}^d$, one form of Stein's identity is

$$\mathbb{E}_p[\nabla_{\boldsymbol{x}} \log p(\boldsymbol{x})^\top \boldsymbol{\phi}(\boldsymbol{x}) + \nabla_{\boldsymbol{x}}^\top \boldsymbol{\phi}(\boldsymbol{x})] = 0, \quad \forall \boldsymbol{\phi},$$

which holds for any differentiable, $\mathbb{R}^d$-valued function $\boldsymbol{\phi}$ that satisfies a proper zero-boundary condition. Stein's identity can be proved by a simple exercise of integration by parts. We may write Stein's identity in a more compact way by defining a Stein operator $\mathcal{P}_{\boldsymbol{x}}$:

$$\mathbb{E}_p[\mathcal{P}_{\boldsymbol{x}}^\top \boldsymbol{\phi}(\boldsymbol{x})] = 0, \quad \text{where} \quad \mathcal{P}_{\boldsymbol{x}}^\top \boldsymbol{\phi}(\boldsymbol{x}) = \nabla_{\boldsymbol{x}} \log p(\boldsymbol{x})^\top \boldsymbol{\phi}(\boldsymbol{x}) + \nabla_{\boldsymbol{x}}^\top \boldsymbol{\phi}(\boldsymbol{x}),$$

where $\mathcal{P}_{\boldsymbol{x}}$ is formally viewed as a $d$-dimensinoal column vector like $\nabla_{\boldsymbol{x}}$, and hence $\mathcal{P}_{\boldsymbol{x}}^\top \boldsymbol{\phi}$ is the inner product of $\mathcal{P}_{\boldsymbol{x}}$ and $\boldsymbol{\phi}$, yielding a scalar-valued function.

The power of Stein's identity is that, for a given distribution $p$, it defines an *infinite* number of functions of form $\mathcal{P}_{\boldsymbol{x}}^\top \boldsymbol{\phi}$ that has zero expectation under $p$, all of which only depend on $p$ through the Stein operator $\mathcal{P}_{\boldsymbol{x}}$, or the score function $\nabla_{\boldsymbol{x}} \log p(\boldsymbol{x}) = \frac{\nabla p(\boldsymbol{x})}{p(\boldsymbol{x})}$, which is independent of the normalization constant in $p$ that is often difficult to calculate.

**Stein Discrepancy on RKHS** Stein's identity can be leveraged to characterize the discrepancy between different distributions. The idea is that, for two different distributions $p \neq q$, there shall exist a function $\boldsymbol{\phi}$ such that $\mathbb{E}_q[\mathcal{P}_{\boldsymbol{x}}^\top \boldsymbol{\phi}] \neq 0$. Consider functions $\boldsymbol{\phi}$ in a $\mathbb{R}^d$-valued reproducing kernel Hilbert space (RKHS) of form $\mathcal{H} = \mathcal{H}_0 \times \cdots \mathcal{H}_0$ where $\mathcal{H}_0$ is a $\mathbb{R}$-valued RKHS with positive definite kernel $k(\boldsymbol{x}, \boldsymbol{x}')$. We may define a *kernelized Stein discrepancy* (KSD) (Liu et al., 2016; Chwialkowski et al., 2016; Oates et al., 2017):

$$\mathbb{D}_k(q \,||\, p) = \max_{\boldsymbol{\phi} \in \mathcal{H}} \left\{ \mathbb{E}_q[\mathcal{P}_{\boldsymbol{x}}^\top \boldsymbol{\phi}(\boldsymbol{x})] \; : \; ||\boldsymbol{\phi}||_{\mathcal{H}} \leq 1 \right\}, \tag{1}$$

The optimal $\boldsymbol{\phi}$ in (1) can be solved in closed form:

$$\boldsymbol{\phi}_{q,p}^*(\cdot) \propto \mathbb{E}_{\boldsymbol{x} \sim q}[\mathcal{P}_{\boldsymbol{x}} k(\boldsymbol{x}, \cdot)], \tag{2}$$

which yields a simple kernel-based representation of KSD:

$$\mathbb{D}_k^2(q \,||\, p) = \mathbb{E}_{\boldsymbol{x}, \boldsymbol{x}' \sim q}[\kappa_p(\boldsymbol{x}, \boldsymbol{x}')], \quad \text{with} \quad \kappa_p(\boldsymbol{x}, \boldsymbol{x}') = \mathcal{P}_{\boldsymbol{x}}^\top (\mathcal{P}_{\boldsymbol{x}'} k(\boldsymbol{x}, \boldsymbol{x}')), \tag{3}$$

where $\boldsymbol{x}$ and $\boldsymbol{x}'$ are i.i.d. draws from $q$, and $\kappa_p(\boldsymbol{x}, \boldsymbol{x}')$ is a new "Steinalized" positive definite kernel obtained by applying the Stein operator twice, first w.r.t. variable $\boldsymbol{x}$ and then $\boldsymbol{x}'$. It turns out that the RKHS related to kernel $\kappa_p(\boldsymbol{x}, \boldsymbol{x}')$ is exactly the space of functions obtained by applying Stein operator on functions in $\mathcal{H}$, that is,

$$\mathcal{H}_p = \{\mathcal{P}_{\boldsymbol{x}}^\top \boldsymbol{\phi} : \forall \boldsymbol{\phi} \in \mathcal{H}\}.$$

By Stein's identity, all the functions in $\mathcal{H}_p$ have zero expectation under $p$. We can also define $\mathcal{H}_p^+$ to be the space of functions in $\mathcal{H}_p$ adding arbitrary constants, that is, $\mathcal{H}_p^+ := \{f(\boldsymbol{x}) + c \colon f \in \mathcal{H}_p, \; c \in \mathbb{R}\}$, which can also be viewed as a RKHS, with kernel $\kappa_p(\boldsymbol{x}, \boldsymbol{x}') + 1$. Stein discrepancy can be viewed as a maximum mean discrepancy (MMD) on the Steinalized RKHS $\mathcal{H}_p^+$ (or equivalently $\mathcal{H}_p$):

$$\mathbb{D}_k(q \,||\, p) = \max_{f \in \mathcal{H}_p^+} \left\{ \mathbb{E}_q f - \mathbb{E}_p f \colon \quad ||f||_{\mathcal{H}_p^+} \leq 1 \right\}. \tag{4}$$

Different from typical MMD, here the RKHS space depends on distribution $p$. In order to make Stein discrepancy discriminative, in that $\mathbb{D}_k(q \,||\, p) = 0$ implies $q = p$, we need to take kernels $k(\boldsymbol{x}, \boldsymbol{x}')$ so that $\mathcal{H}_p^+$ is sufficiently large. It has been shown that this can be achieved if $k(\boldsymbol{x}, \boldsymbol{x}')$ is strictly positive definite or universal, in a proper technical sense (Liu et al., 2016; Chwialkowski et al., 2016; Gorham & Mackey, 2017; Oates et al., 2017).

It is useful to consider the kernels in a random feature representation (Rahimi & Recht, 2007),

$$k(\boldsymbol{x}, \boldsymbol{x}') = \mathbb{E}_{\boldsymbol{w} \sim p_{\boldsymbol{w}}}[\phi(\boldsymbol{x}, \boldsymbol{w}) \phi(\boldsymbol{x}', \boldsymbol{w})], \tag{5}$$

where $\phi(\boldsymbol{x}, \boldsymbol{w})$ is a set of features indexed by a random parameter $\boldsymbol{w}$ drawn from a distribution $p_{\boldsymbol{w}}$. For example, the Gaussian RBF kernel $k(\boldsymbol{x}, \boldsymbol{x}') = \exp(-\frac{1}{2h^2}||\boldsymbol{x} - \boldsymbol{x}'||_2^2)$ admits

$$\phi(\boldsymbol{x}, \boldsymbol{w}) = \sqrt{2} \cos(\frac{1}{h} \boldsymbol{w}_1^\top \boldsymbol{x} + w_0), \tag{6}$$

where $w_0 \sim \text{Unif}([0, 2\pi])$ and $\boldsymbol{w}_1 \sim \mathcal{N}(0, I)$. With the random feature representation, KSD can be rewritten into

$$\mathbb{D}_k^2(q \,||\, p) = \mathbb{E}_{\boldsymbol{w} \sim p_{\boldsymbol{w}}}\left[||\mathbb{E}_{\boldsymbol{x} \sim q}[\mathcal{P}_{\boldsymbol{x}}\phi(\boldsymbol{x}, \boldsymbol{w})]||^2\right], \tag{7}$$

which can be viewed as the mean square error of Stein's identity $\mathbb{E}_{\boldsymbol{x} \sim q}[\mathcal{P}_{\boldsymbol{x}}\phi(\boldsymbol{x}, \boldsymbol{w})] = 0$ over the random features. $\mathbb{D}_k^2(q \,||\, p) = 0$ shall imply $q = p$ if the feature set $\mathcal{G} = \{\phi(\boldsymbol{x}, \boldsymbol{w}) \colon \forall \boldsymbol{w}\}$ is rich enough. Note that the RKHS $\mathcal{H}$ and feature set $\mathcal{G}$ are different; Stein discrepancy is an *expected* loss function on $\mathcal{G}$ as shown in (7), but a *worst-case* loss on $\mathcal{H}$ as shown in (1).

**Stein Variational Gradient Descent (SVGD)**    SVGD is a deterministic sampling algorithm motivated by Stein discrepancy. It is based on the following basic observation: given a distribution $q$, assume $q_{[\phi]}$ is the distribution of $\boldsymbol{x}' = \boldsymbol{x} + \epsilon\phi(\boldsymbol{x})$ obtained by updating $\boldsymbol{x}$ with a velocity field $\phi$, where $\epsilon$ is a small step size, then we have

$$\text{KL}(q_{[\phi]} \,||\, p) = \text{KL}(q \,||\, p) - \epsilon\mathbb{E}_q[\mathcal{P}_{\boldsymbol{x}}^\top\phi] + O(\epsilon^2),$$

which shows that the decrease of KL divergence is dominated by $\epsilon\mathbb{E}_q[\mathcal{P}_{\boldsymbol{x}}^\top\phi]$. In order to choose $\phi$ to make $q_{[\phi]}$ move towards $p$ as fast as possible, we should choose $\phi$ to maximize $\mathbb{E}_q[\mathcal{P}_{\boldsymbol{x}}^\top\phi]$, whose solution is exactly $\phi_{q,p}^*(\cdot) \propto \mathbb{E}_{\boldsymbol{x} \sim q}[\mathcal{P}_{\boldsymbol{x}}k(\boldsymbol{x}, \cdot)]$ as shown in (2). This suggests that $\phi_{q,p}^*$ happens to be the best velocity field that pushes the probability mass of $q$ towards $p$ as fast as possible.

Motivated by this, SVGD approximates $q$ with the empirical distribution of a set of particles $\{\boldsymbol{x}_i\}_{i=1}^n$, and iteratively updates the particles by

$$\boldsymbol{x}_i \leftarrow \boldsymbol{x}_i + \frac{\epsilon}{n}\sum_{j=1}^n[\mathcal{P}_{\boldsymbol{x}_j}k(\boldsymbol{x}_j, \boldsymbol{x}_i)]. \tag{8}$$

Liu et al. (2017) studied the asymptotic properties of the dynamic system underlying SVGD, showing that the evolution of the limit density of the particles when $n \to \infty$ can be captured by a nonlinear Fokker-Planck equation, and established its weak convergence to the target distribution $p$.

However, the analysis in Liu et al. (2017) and Lu et al. (2018) do not cover the case when the sample size $n$ is finite, which is more relevant to the practical performance. We address this problem by directly analyzing the properties of the fixed point equation of SVGD, yielding results that work for finite sample size $n$, also independent of the update rule used to arrive the fixed points.

## 3    SVGD as Moment Matching

This section presents our main results on the moment matching properties of SVGD and the related Stein matching sets. We start with Section 3.1 which introduces the basic idea and characterizes the Stein matching set of SVGD with general positive definite kernels. We then analyze in Section 3.2 the special case when the rank of the kernel is less than the particle size, in which case the Stein matching set is independent of the fixed points themselves. Section 3.3 shows that SVGD with linear features exactly estimates the first two second-order moments of Gaussian distributions. Section 3.4 establishes a probabilistic bound when random features are used.

### 3.1    Fixed Point of SVGD

Our basic idea is rather simple to illustrate. Assume $X^* = \{\boldsymbol{x}_i^*\}_{i=1}^n$ is the fixed point of SVGD and $\hat{\mu}_{X^*}$ its related empirical measure, then according to (8), the fixed point condition of SVGD ensures

$$\mathbb{E}_{\boldsymbol{x} \sim \hat{\mu}_{X^*}}[\mathcal{P}_{\boldsymbol{x}}k(\boldsymbol{x}, \boldsymbol{x}_i^*)] = 0, \qquad \forall i = 1, \ldots, n. \tag{9}$$

On the other hand, by Stein's identity, we have

$$\mathbb{E}_{\boldsymbol{x} \sim p}[\mathcal{P}_{\boldsymbol{x}}k(\boldsymbol{x}, \boldsymbol{x}_i^*)] = 0, \qquad \forall i = 1, \ldots, n.$$

This suggests that $\hat{\mu}_{X^*}$ *exactly* estimates the expectation of functions of form $f(\boldsymbol{x}) = \mathcal{P}_{\boldsymbol{x}}k(\boldsymbol{x}, \boldsymbol{x}_i^*)$ under $p$, all of which are zero. By the linearity of expectation, the same holds for all the functions in the linear span of $\mathcal{P}_{\boldsymbol{x}}k(\boldsymbol{x}, \boldsymbol{x}_i^*)$.

**Lemma 3.1.** *Assume $X^* = \{\boldsymbol{x}_i^*\}_{i=1}^n$ satisfies the fixed point equation (9) of SVGD. We have*

$$\frac{1}{n}\sum_{i=1}^n f(\boldsymbol{x}_i^*) = \mathbb{E}_p f, \qquad \forall f \in \mathcal{F}^*,$$

*where the Stein matching set $\mathcal{F}^*$ is the linear span of $\{\mathcal{P}_{\boldsymbol{x}}k(\boldsymbol{x}, \boldsymbol{x}_i^*)\}_{i=1}^n \cup \{1\}$, that is, $\mathcal{F}^*$ consists of*

$$f(\boldsymbol{x}) = \sum_{i=1}^n \boldsymbol{a}_i^\top \mathcal{P}_{\boldsymbol{x}}k(\boldsymbol{x}, \boldsymbol{x}_i^*) \;+\; b, \quad \forall \boldsymbol{a}_i \in \mathbb{R}^d,\; b \in \mathbb{R}.$$

*Equivalently, $f(\boldsymbol{x}) = \mathcal{P}_{\boldsymbol{x}}^\top \boldsymbol{\phi}(\boldsymbol{x}) + b$ and $\boldsymbol{\phi}$ is in the linear span of $\{k(\boldsymbol{x}, \boldsymbol{x}_i^*)\}_{i=1}^n$, that is, $\boldsymbol{\phi}(\boldsymbol{x}) = \sum_{i=1}^n \boldsymbol{a}_i k(\boldsymbol{x}, \boldsymbol{x}_i^*)$.*

Extending Lemma 3.1, one can readily see that the SVGD fixed points can approximate the expectation of functions that are close to $\mathcal{F}^*$. Specifically, let $\mathcal{F}_\epsilon^*$ be the $\epsilon$ neighborhood of $\mathcal{F}^*$, that is, $\mathcal{F}_\epsilon^* = \{f \colon \inf_{f' \in \mathcal{F}} ||f - f'||_\infty \le \epsilon\}$, then it is easily shown that

$$\left| \frac{1}{n}\sum_{i=1}^n f(\boldsymbol{x}_i^*) - \mathbb{E}_p f \right| \le 2\epsilon, \qquad \forall f \in \mathcal{F}_\epsilon^*.$$

Therefore, the SVGD approximation can be viewed as *prioritizing* the functions within, or close to, $\mathcal{F}^*$. This is different in nature from Monte Carlo, which approximates the expectation of *all bounded variance functions* with the same $O(1/\sqrt{n})$ error rate. Instead, SVGD shares more similarity with the *quadrature* and *sigma point methods*, which also find points (particles) to match the expectation on certain class of functions, but mostly only on polynomial functions and for simple distributions such as uniform or Gaussian distributions. SVGD provides a more general approach that can match moments of richer classes of functions for more general complex multivariate distributions. As we show in Section 3.3, when using polynomial kernels, SVGD reduces to matching polynomials when applied to multivariate Gaussian distributions.

In this view, the performance of SVGD is essentially decided by the Stein matching set $\mathcal{F}^*$. We shall design the algorithm, by engineering the kernels or feature maps, to make $\mathcal{F}^*$ as large as possible in order to approximate the distribution well, or include the test functions of actual interest, such as mean and variance.

## 3.2   Fixed Point of Feature-based SVGD

One undesirable property of $\mathcal{F}^*$ in Lemma 3.1 is that it depends on the values of the fixed point particles $X^*$, whose properties are difficult to characterize *a priori*. This makes it difficult to infer what kernel should be used to obtain a desirable $\mathcal{F}^*$. It turns out the dependency of $\mathcal{F}$ on $X^*$ can be essentially decoupled by using *degenerated kernels* corresponding to a finite number of feature maps. Specifically, we consider kernels of form

$$k(\boldsymbol{x}, \boldsymbol{x}') = \sum_{\ell=1}^m \phi_\ell(\boldsymbol{x})\phi_\ell(\boldsymbol{x}'),$$

where we assume the number $m$ of features is no larger than the particle size $n$. Then, the fixed point of SVGD reduces to

$$\mathbb{E}_{\boldsymbol{x}\sim\hat{\mu}_{X^*}}\Big[\sum_{\ell=1}^m \mathcal{P}_{\boldsymbol{x}}\phi_\ell(\boldsymbol{x})\phi_\ell(\boldsymbol{x}_j^*)\Big] = 0, \quad \forall j \in [n]. \tag{10}$$

Define $\Phi = [\phi_\ell(\boldsymbol{x}_j^*)]_{\ell,j}$ which is a matrix of size $(m \times n)$. If $rank(\Phi) \ge m$, then (10) reduces to

$$\mathbb{E}_{\boldsymbol{x}\sim\hat{\mu}_{X^*}}[\mathcal{P}_{\boldsymbol{x}}\phi_\ell(\boldsymbol{x})] = 0, \quad \forall \ell = 1, \ldots, m, \tag{11}$$

where the test function $f(\boldsymbol{x}) := \mathcal{P}_{\boldsymbol{x}}\phi_\ell(\boldsymbol{x})$ no longer depends on the fixed point $X^*$.

**Theorem 3.2.** *Assume $X^*$ is a fixed point of SVGD with kernel $k(\boldsymbol{x}, \boldsymbol{x}') = \sum_{\ell=1}^m \phi_\ell(\boldsymbol{x})\phi_\ell(\boldsymbol{x}')$. Define the $(m \times n)$ matrix $\Phi = [\phi_\ell(\boldsymbol{x}_i^*)]_{\ell\in[m], i\in[n]}$. If $\mathrm{rank}(\Phi) \ge m$, then*

$$\frac{1}{n}\sum_{i=1}^n f(\boldsymbol{x}_i^*) = \mathbb{E}_p f, \quad \forall f \in \mathcal{F}^*,$$

*where the Stein matching set $\mathcal{F}^*$ is the linear span of $\{\mathcal{P}_{\boldsymbol{x}}\phi_\ell(\boldsymbol{x})\}_{\ell=1}^m \cup \{1\}$, that is, it is set of the functions of form*

$$f(\boldsymbol{x}) = \sum_{\ell=1}^m \boldsymbol{a}_\ell^\top \mathcal{P}_{\boldsymbol{x}}\phi_\ell(\boldsymbol{x}) \; + \; b, \qquad\qquad \forall \, \boldsymbol{a}_\ell \in \mathbb{R}^d, \; b \in \mathbb{R}. \qquad (12)$$

Note that the rank condition implies that we must have $m \le n$. The idea is that $n$ particles can at most match $n$ linearly independent features exactly. Here, although the rank condition still depends on the fixed point $X^* = \{\boldsymbol{x}_i^*\}_{i=1}^n$ and cannot be guaranteed *a priori*, it can be numerically verified once we obtain the values of $X^*$. In our experiments, we find that the rank condition tends to always hold practically when $n = m$. In cases when it does fail to satisfy, we can always rerun the algorithm with a larger $n$ until it is satisfied. Intuitively, it seems to require bad luck to have $\Phi$ low rank when there are more particles than features ($n \ge m$), although a theoretical guarantee is still missing.

**Query-Specific Inference as Solving Stein Equation**   Assume we are interested in a *query-specific* task of estimating $\mathbb{E}_p f$ for a specific test function $f$. In this case, we should ideally select the features $\{\phi_\ell\}_\ell$ such that (12) holds to yield an exact estimation of $\mathbb{E}_p f$. By the linearity of the Stein operator, (12) is equivalent to

$$\texttt{Stein Equation:} \qquad\qquad f(\boldsymbol{x}) = \mathcal{P}_{\boldsymbol{x}}^\top \boldsymbol{\phi}(\boldsymbol{x}) + b, \qquad (13)$$

where $\boldsymbol{\phi}(\boldsymbol{x}) = \sum_{\ell=1}^m \boldsymbol{a}_\ell \phi_\ell(\boldsymbol{x})$. Eq (13) is known as *Stein Equation* when solving $\boldsymbol{\phi}$ and $b$ with a given $f$, which effectively calculates the inverse of Stein operator.

Stein equation plays a central role in Stein's method as a theoretical tool (Barbour & Chen, 2005). Here, we highlight its fundamental connection to the approximate inference problem: if we can exactly solve $\boldsymbol{\phi}$ and $b$ for a given $f$, then the inference problem regarding $f$ is already solved (without running SVGD), since we can easily see that $\mathbb{E}_p f = b$ by taking expectation from both sides of (13).

Mathematically, this reduces the integration problem of estimating $\mathbb{E}_p f$ into solving a differential equation. It suggests that Stein equation is at least as hard as the inference problem itself, and we should not expect a tractable way to solve it in general cases. On the other hand, it suggested that efficient ways of approximate inference may be developed by approximate solutions of Stein equation. Similar idea has been investigated in Oates et al. (2017), which developed a kernel approximation of Stein equation in the case based on a given set of points. SVGD allows us to further extend this idea by optimizing the set of points (particles) on which approximation is defined.

## 3.3   Linear Feature SVGD is Exact for Gaussian

Although Stein equation is difficult to solve in general, it is significantly simplified when the distribution $p$ of interest is Gaussian. In the following, we show that when $p$ is a multivariate Gaussian distribution, we can use linear features, relating to a linear kernel $k(\boldsymbol{x}, \boldsymbol{x}') = \boldsymbol{x}^\top \boldsymbol{x}' + 1$, to ensure that SVGD exactly estimates all the first and second order moments of $p$. This insight provides an important practical guidance on the optimal kernel choices for Gaussian-like distributions.

**Theorem 3.3.** *Assume $X^*$ is a fixed point of SVGD with polynomial kernel $k(\boldsymbol{x}, \boldsymbol{x}') = \boldsymbol{x}^\top \boldsymbol{x}' + 1$. Let $\mathcal{F}^*$ be the Stein matching set in Theorem 3.2. If $p$ is a multivariate normal distribution on $\mathbb{R}^d$, then $\mathcal{F} \subseteq \mathrm{Poly}(2)$, where $\mathrm{Poly}(2)$ is the set of all polynomials upto the second order, that is, $\mathrm{Poly}(2) = \{\boldsymbol{x}^\top A \boldsymbol{x} + \boldsymbol{b}^\top \boldsymbol{x} + c \colon A \in \mathbb{R}^{d \times d}, \; \boldsymbol{b} \in \mathbb{R}^d, \; c \in \mathbb{R}\}$.*

*Further, denote by $\Phi$ the $(d+1) \times n$ matrix defined by*

$$\Phi = \begin{bmatrix} \boldsymbol{x}_1 & \boldsymbol{x}_2 & \cdots & \boldsymbol{x}_n \\ 1 & 1 & \cdots & 1 \end{bmatrix}.$$

*If $\mathrm{rank}(\Phi) \ge d + 1$, then $\mathcal{F} = \mathrm{Poly}(2)$. In this case, any fixed point of SVGD exactly estimates both the mean and the covariance matrix of the target distribution.*

More generally, if the features are polynomials of order $j$, its related Stein matching set should be polynomials of order $j + 1$ for Gaussian distributions. We do not investigate this further because it is less common to estimate higher order moments in multivariate settings.

Theorem 3.3 suggests that it is a good heuristic to include linear features in SVGD, because Gaussian-like distributions appear widely thanks to the central limit theorem and Bernstein–von Mises theorem, and the main goal of inference is often to estimate the mean and variance. In contrast, the more commonly used Gaussian RBF kernel does not have similar exact recovery results for the mean and variance, even for Gaussian distributions.

A nice property of our result is that once we use fewer features than the particles and solve the fixed point exactly, the features do not "interfere" with each other. This allows us to "program" our algorithm by adding different types of features that serve different purposes in different cases.

## 3.4 Random feature SVGD

The linear features are not sufficient for providing the consistent estimation of the whole distribution, even for Gaussian distributions. Non-degenerate kernels are required to obtain bounds on the whole distributions, but they complicate the analysis because their Stein matching set depends on the solution $X^*$ as shown in Lemma C.1. Random features can be used to sidestep this difficulty (Rahimi & Recht, 2007), enabling us to analyze a random feature variant of SVGD with probabilistic bounds.

To set up, assume $k(\boldsymbol{x}, \boldsymbol{x}')$ is a universal kernel whose Stein discrepancy $\mathbb{D}_k(q \,||\, p)$ yields a discriminative measure of differences between distributions. Assume $k(\boldsymbol{x}, \boldsymbol{x}')$ yields the random feature representation in (5), and we can approximate it by drawing $m$ random features,

$$\hat{k}(\boldsymbol{x}, \boldsymbol{x}') = \frac{1}{m} \sum_{\ell=1}^{m} \phi(\boldsymbol{x}, \boldsymbol{w}_\ell) \phi(\boldsymbol{x}', \boldsymbol{w}_\ell),$$

where $\boldsymbol{w}_\ell$ are i.i.d. drawn from $p_{\boldsymbol{w}}$. We assume $m \le n$, then running SVGD with kernel $\hat{k}(\boldsymbol{x}, \boldsymbol{x}')$ (with the random features fixed during the iterations) yields a matching set that decouples with the fixed point $X^*$. In this way, our result below establish that $\mathbb{D}_k(\hat{\mu}_{X^*} \,||\, p) = \tilde{O}(1/\sqrt{n})$ with high probability. According to (4), this provides a uniform bound of $\mathbb{E}_{\hat{\mu}_{X^*}} f - \mathbb{E}_p f$ for all functions in the unit ball of $\mathcal{H}_p^+$.

Here, random features are introduced mainly for facilitating theoretical analysis, but we also find random feature SVGD works comparably, and sometimes even better than SVGD with the original non-degenerate kernel (see Appendix). This is because with a finite number $n$ of particles, at most $n$ function basis of $k(x, x')$ can be effectively used, even if $k(x, x')$ itself has an infinite rank. From the perspective of moment matching, there is no benefit to use universal kernels when the particle size $n$ is finite.

In the sequel, we first explain the intuitive idea behind our result, highlighting a perspective that views inference as fitting a zero-valued curve with Stein's identity, and then introduce technical details.

**Distributional Inference as Fitting Stein's Identity** Recall that our goal can be viewed as finding particles $X^* = \{\boldsymbol{x}_i^*\}$ such that their empirical $\hat{\mu}_{X^*}$ approximates the target distribution $p$. We re-frame this into finding $\hat{\mu}_{X^*}$ such that Stein's identity holds (approximately):

$$\text{Find } \hat{\mu}_{X^*} \qquad s.t. \qquad \mathbb{E}_{\hat{\mu}_{X^*}}[\mathcal{P}_{\boldsymbol{x}} \phi(\boldsymbol{x}, \boldsymbol{w})] \approx 0, \quad \forall \boldsymbol{w}.$$

We may view this as a special curve fitting problem: considering $\boldsymbol{g}_X(\boldsymbol{w}) = \mathbb{E}_{\hat{\mu}_X}[\mathcal{P}_{\boldsymbol{x}} \phi(\boldsymbol{x}, \boldsymbol{w})]$, we want to find "parameter" $X$ such that $\boldsymbol{g}_X(\boldsymbol{w}) \approx 0$ for all inputs $\boldsymbol{w}$. The kernelized Stein discrepancy (KSD), as shown in (7), can be viewed as the expected rooted mean square loss of this fitting problem:

$$\mathbb{D}_k^2(\hat{\mu}_X \,||\, p) = \mathbb{E}_{\boldsymbol{w} \sim p_{\boldsymbol{w}}} \left[ ||\boldsymbol{g}_X(\boldsymbol{w})||_2^2 \right] \qquad (14)$$

When replacing $k(\boldsymbol{x}, \boldsymbol{x}')$ with its random feature approximation $\hat{k}(\boldsymbol{x}, \boldsymbol{x}')$, the corresponding KSD can be viewed as an empirical loss on random sample $\{\boldsymbol{w}_\ell\}$ from $p_{\boldsymbol{w}}$:

$$\mathbb{D}_{\hat{k}}^2(\hat{\mu}_X \,||\, p) = \frac{1}{m} \sum_{\ell=1}^{m} \left[ ||\boldsymbol{g}_X(\boldsymbol{w}_\ell)||_2^2 \right].$$

By running SVGD with $\hat{k}(\boldsymbol{x}, \boldsymbol{x}')$, we acheive $\boldsymbol{g}_{X^*}(\boldsymbol{w}_\ell) = 0$ for all $\ell$ at the fixed point, implying a zero empirical loss $\mathbb{D}_{\hat{k}}(\hat{\mu}_{X^*} \,||\, p) = 0$ assuming the rank condition holds.

The key question, however, is to bound the expected loss $\mathbb{D}_k(\hat{\mu}_{X^*} \,||\, p)$, which can be achieved using generalization bounds in statistical learning theory. In fact, standard results in learning theory

suggests that the difference between the empirical loss and expected loss is $O(m^{-1/2})$, yielding $\mathbb{D}_k^2(\hat{\mu}_{X^*} \parallel p) = O(m^{-1/2})$. However, following (4), this implies $\mathbb{E}_{\hat{\mu}_{X^*}} f - \mathbb{E}_p f = O(m^{-1/4})$ for $f \in \mathcal{H}_p^+$, which does not acheive the standard $O(m^{-1/2})$. Fortunately, note that our setting is noise-free, and we achieve zero empirical loss; thus, we can get a better rate of $\mathbb{D}_k^2(\hat{\mu}_X \parallel p) = \tilde{O}(m^{-1})$ using the techniques in Srebro et al. (2010).

**Bound for Random Features** We now present our concentration bounds of random feature SVGD.

**Assumption 3.4.** *1) Assume $\{\phi(x, w_\ell)\}_{\ell=1}^m$ is a set of random features with $w_\ell$ i.i.d. drawn from $p_w$ on domain $\mathcal{W}$, and $X^* = \{x_i^*\}_{i=1}^n$ is an approximate fixed point of SVGD with random feature $\phi(x, w_\ell)$ in the sense that*

$$|\mathbb{E}_{x \sim \hat{\mu}_{X^*}} \mathcal{P}_{x^j} \phi(x, w_\ell)| \leq \frac{\epsilon_j}{\sqrt{m}}.$$

*where $\mathcal{P}_{x^j}$ is the Stein operator w.r.t. the $j$-th coordinate $x^j$ of $x$. Assume $\epsilon^2 := \sum_{j=1}^d \epsilon_j^2 < \infty$.*

*2) Let $\sup_{x \in \mathcal{X}, w \in \mathcal{W}} |\mathcal{P}_{x^j} \phi(x, w)| = M_j$, and $M^2 := \sum_{j=1}^d M_j^2 < \infty$. This may imply that $\mathcal{X}$ has to be compact since $\nabla_x \log p(x)$ is typically unbounded on non-compact $\mathcal{X}$ (e.g., when $p$ is standard Gaussian, $\nabla_x \log p(x) = x$).*

*3) Define function set*

$$\mathcal{P}_j \Phi = \{w \mapsto \mathcal{P}_{x^j} \phi(x, w) \colon \forall x \in \mathcal{X}\}.$$

*We assume the Rademacher complexity of $\mathcal{P}_j \Phi$ satisfies $\mathcal{R}_m(\mathcal{P}_j \Phi) \leq R_j / \sqrt{m}$, and $R^2 := \sum_{j=1}^d R_j^2 < \infty$.*

**Theorem 3.5.** *Under Assumption 3.4, for any $\delta > 0$, we have with at least probability $1 - \delta$ (in terms of the randomness of feature parameters $\{w_\ell\}_{\ell=1}^m$),*

$$\mathbb{D}_k(\hat{\mu}_{X^*} \parallel p) \leq \frac{C}{\sqrt{m}} \left[\epsilon^2 + \log^3 m + \log(1/\delta)\right]^{1/2}, \tag{15}$$

*where $C$ is a constant that depends on $R$ and $M$.*

**Remark** Recalling (4), Eq (15) provides a uniform bound

$$\sup_{\|f\|_{\mathcal{H}_p^+} \leq 1} \left\{\mathbb{E}_{\mu_{X^*}} f - \mathbb{E}_p f\right\} = O(m^{-1/2} \log^{1.5} m).$$

This is a uniform bound that controls the worse error uniformly among all $f \in \mathcal{H}_p^+$. It is unclear if the logarithm factor $\log^{1.5} m$ is essential. In the following, we present a result that has an $O(1/\sqrt{m})$ rate, without the logarithm factor, but only holds for individual functions.

**Theorem 3.6.** *Let $\mathcal{F}_\infty$ be the set of linear span of the Steinalized features:*

$$f(x) = \mathbb{E}_{w \sim p_w}[v(w)^\top \mathcal{P}_x \phi(x, w)], \tag{16}$$

*where $v(w) = [v_1(w), \ldots, v_d(w)] \in \mathbb{R}^d$ is the combination weights that satisfy $\sup_w \|v(w)\|_\infty < \infty$. We may define a norm on $\mathcal{F}_\infty$ by $\|f\|_{\mathcal{F}_\infty}^2 := \inf_v \sum_{j=1}^d \sup_w |v_j(w)|^2$, where $\inf_v$ is taken on all $v(w)$ that satisfies (16).*

*Assume Assumption 3.4 holds, then for any given function $f \in \mathcal{F}_\infty$ with $\|f\|_{\mathcal{F}_\infty} \leq 1$, we have with at least probability $1 - \delta$,*

$$|\mathbb{E}_{\hat{\mu}_{X^*}} f - \mathbb{E}_p f| \leq \frac{C}{\sqrt{m}}(1 + \epsilon + \sqrt{2 \log(1/\delta)}),$$

*where $C$ is a constant that depends on $R$ and $M$.*

The $\mathcal{F}_\infty$ defined above is closely related to the RKHS $\mathcal{H}_p$. In fact, one can show that $\mathcal{F}_\infty$ is a dense subset of $\mathcal{H}_p$ (Rahimi & Recht, 2008) and is hence quite rich if $k(x, x')$ is set to be universal.

# 4 Conclusion

We analyze SVGD through the eyes of moment matching. Our results are non-asymptotic in nature and provide an insightful framework for understanding the influence of kernels in the behavior of SVGD fixed points. Our framework suggests promising directions to develop systematic ways of optimizing the choice of kernels, especially for the query-specific inference that focuses on specific test functions. A particularly appealing idea is to "program" the inference algorithm by adding features that serve specific purposes so that the algorithm can be easily adapted to meet the needs of different users. In general, we expect that the connection between approximation inference and Stein's identity and Stein equation will provide further opportunities for deriving new generations of approximate inference algorithms.

Another advantage of our framework is that it separates the design of the fixed point equation with the numerical algorithm used to achieve the fixed point. In this way, the iterative algorithm does not have to be derived as an approximation of an infinite dimensional gradient flow, in contrast to the original SVGD. This allows us to apply various practical numerical methods and acceleration techniques to solve the fixed point equation faster, with convergence guarantees.

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
