[Supplementary Material]

# A Proof of Theorem 3.3

*Proof.* Assume $p$ is multivariate normal $\mathcal{N}(\boldsymbol{\mu}, Q^{-1})$ where $Q$ is the inverse covariance matrix. We have $\nabla_{\boldsymbol{x}} \log p(\boldsymbol{x}) = -Q(\boldsymbol{x} - \boldsymbol{\mu})$. Since $k(\boldsymbol{x}, \boldsymbol{x}') = \boldsymbol{x}^\top \boldsymbol{x}' + 1$, the functions in $\mathcal{F}^*$ should have a form of

$$f(\boldsymbol{x}) = \sum_{i=1}^n \boldsymbol{a}_i^\top [-Q(\boldsymbol{x} - \boldsymbol{\mu})(\boldsymbol{x}_i^\top \boldsymbol{x} + 1) + \boldsymbol{x}_i] + b$$
$$= \boldsymbol{x}^\top W \boldsymbol{x} + \boldsymbol{v}^\top \boldsymbol{x} + c,$$

where

$$W = -\sum_{i=1}^n \boldsymbol{x}_i \boldsymbol{a}_i^\top Q,$$

$$\boldsymbol{v} = \sum_{i=1}^n (\boldsymbol{\mu}^\top \boldsymbol{x}_i - 1) Q \boldsymbol{a}_i$$

$$c = b + \sum_{i=1}^n \boldsymbol{a}_i^\top (Q\boldsymbol{\mu} + \boldsymbol{x}_i).$$

Denote by $X = [\boldsymbol{x}_1, \ldots, \boldsymbol{x}_n]$ the $(d \times n)$ matrix, $A = [\boldsymbol{a}_1, \ldots, \boldsymbol{a}_n]$ the $(d \times n)$ matrix, and $B = QA$. We have

$$W = -XB^\top, \tag{17}$$
$$\boldsymbol{v}^\top = (\boldsymbol{\mu}^\top X - \boldsymbol{e}^\top) B^\top, \tag{18}$$
$$c = b + \boldsymbol{e}^\top B^\top \boldsymbol{\mu} + \operatorname{tr}(XA^\top), \tag{19}$$

where $\boldsymbol{e}$ is the $\mathbb{R}^d$-vector of all ones. Eq. (17) and (18) are equivalent to

$$\begin{bmatrix} -X \\ \boldsymbol{\mu}^\top X - \boldsymbol{e} \end{bmatrix} B = \begin{bmatrix} W \\ \boldsymbol{v}^\top \end{bmatrix} \tag{20}$$

We just need to show that for any value of $W \in \mathbb{R}^{d \times d}$, $\boldsymbol{v} \in \mathbb{R}^d$ and $c \in \mathbb{R}$ there exists $A = [\boldsymbol{a}_1, \ldots, \boldsymbol{a}_n]$ and $b$ that satisfies the above equation. This is equivalent to

$$\begin{bmatrix} -I, 0 \\ \boldsymbol{\mu}^\top, -1 \end{bmatrix} \Phi B = \begin{bmatrix} W \\ \boldsymbol{v}^\top \end{bmatrix}$$

Since $\begin{bmatrix} -I, 0 \\ \boldsymbol{\mu}^\top, -1 \end{bmatrix}$ is always full rank, if $\Phi$ has a rank at least $d + 1$, then (20) exits a solution for $B$. We can then get $A = Q^{-1} B$ and solve $b$ from (19). $\qquad \square$

# B Proof of Theorem 3.5

*Proof.* A loss function is $H$-smooth iff its derivative is $H$-Lipschitz. For twice differentiable $\phi$, this just means $|\phi''| \leq H$. The following result from Srebro et al. (2010) is key to our proof.

**Theorem B.1** (Srebro et al. (2010) Theorem 1). *For an $H$-smooth non-negative loss $\phi$, such that $\forall_{x,y,h} |\phi(h(x), y)| \leq b$, for any $\delta > 0$, we have with probability at least $1 - \delta$ over a random sample of size $n$ that, for any $h \in \mathcal{H}$, we have*

$$L(h) \leq \hat{L}(h) + K \left[ \sqrt{\hat{L}(h)} \left( \sqrt{H} \log^{1.5} n \mathcal{R}_n(\mathcal{H}) + \sqrt{\frac{b \log(1/\delta)}{n}} \right) + H \log^3 n \mathcal{R}_n^2(\mathcal{H}) + \frac{b \log(1/\delta)}{n} \right].$$

*where $K$ is a numerical constant that satisfies $K < 10^5$.*

We now apply this result to bound kernelized Stein discrepancy. Take $\phi(x, y) = (x - y)^2$, then $H = 2$. Define $g_{X,j}(\boldsymbol{w}) = \mathbb{E}_{\hat{\mu}_X}[\mathcal{P}_{x^j}\phi(\boldsymbol{x}, \boldsymbol{w})]$ and $\mathcal{G}_j = \{g_{X,j} \colon \forall X \in \mathcal{X}^n\}$. Recall that the Stein discrepancy can be viewed as the sum of mean square losses of fitting $g_{X,j}$ to the zero-valued line:

$$\mathbb{D}_k^2(\hat{\mu}_X \parallel p) = \sum_{j=1}^d L_j(g_{X,j}), \qquad \text{where} \qquad L_j(g_{X,j}) = \mathbb{E}_{\boldsymbol{w} \sim p_{\boldsymbol{w}}}[(g_{X,j}(\boldsymbol{w}) - 0)^2],$$

$$\mathbb{D}_{\hat{k}}^2(\hat{\mu}_X \parallel p) = \sum_{j=1}^d \hat{L}_j(g_{X,j}), \qquad \text{where} \qquad \hat{L}_j(g_{X,j}) = \frac{1}{m} \sum_{\ell=1}^m [(g_{X,j}(\boldsymbol{w}_\ell) - 0)^2].$$

We now apply Theorem B.1 to each bound the difference between the expected loss $L_j(g_{X,j})$ and the empirical loss $\hat{L}_j(g_{X,j})$. From Assumption 3.4.2, we have $\sup_{X,\boldsymbol{w}} |g_{X,j}(\boldsymbol{w})| \leq M_j$. This is because

$$|g_{X,j}(\boldsymbol{w})| = |\frac{1}{n} \sum_{i=1}^n \mathcal{P}_{x^j}\phi(\boldsymbol{x}, \boldsymbol{w})| \leq \frac{1}{n} \sum_{i=1}^n |\mathcal{P}_{x^j}\phi(\boldsymbol{x}, \boldsymbol{w})| \leq M_j.$$

Using Theorem B.1, we have with probability $1 - \delta$, for any $X$,

$$L_j(g_{X,j}) \leq \hat{L}_j(g_{X,j}) + K\left[\hat{L}_j(g_{X,j})\left(\sqrt{2}\log^{1.5} m \mathcal{R}_m(\mathcal{G}_j) + \sqrt{\frac{M_j^2 \log(1/\delta)}{m}}\right)\right.$$
$$\left. + 2\log^3 m \mathcal{R}_m^2(\mathcal{G}_j) + \frac{M_j^2 \log(1/\delta)}{m}\right].$$

By Assumption 3.4.1, $|g_{X^*,j}(\boldsymbol{w}_\ell)| \leq \frac{\epsilon_j}{\sqrt{m}}$ for $\forall \ell = 1, \ldots, m$ at the approximate fixed point $X^*$. We have $\hat{L}_j(g_{X,j}) \leq \frac{\epsilon_j}{\sqrt{m}}$. By Assumption 3.4.3, we have $\mathcal{R}_m(\mathcal{G}_j) \leq R_j/\sqrt{m}$. Therefore,

$$L_j(g_{X^*,j}) \leq \frac{\epsilon_j^2}{m} + K\left[\frac{\epsilon_j}{\sqrt{m}}\left(\sqrt{2}\log^{1.5} m \frac{R_j}{\sqrt{m}} + \sqrt{\frac{M_j^2 \log(1/\delta)}{m}}\right) + 2\log^3 m \frac{R_j^2}{m} + \frac{M_j^2 \log(1/\delta)}{m}\right].$$

Summing across $j = 1, \ldots, d$, we get

$$\mathbb{D}_k^2(\hat{\mu}_{X^*} \parallel p)$$
$$\leq \frac{1}{m} \sum_{j=1}^d \left[\epsilon_j^2 + K\left(\sqrt{2}R_j\epsilon_j \log^{1.5} m + M_j\epsilon_j \sqrt{\log(1/\delta)} + 2R_j^2 \log^3 m + M_j^2 \log(1/\delta)\right)\right]$$
$$\leq \frac{1}{m}\left[\epsilon^2 + K\left(\sqrt{2}R\epsilon \log^{1.5} m + M\epsilon \sqrt{\log(1/\delta)} + 2R^2 \log^3 m + M^2 \log(1/\delta)\right)\right]$$
$$\leq \frac{C^2}{m}\left[\epsilon^2 + \log^3 m + \log(1/\delta)\right],$$

where $C^2 = \max\{1 + \frac{1}{\sqrt{2}}KR + \frac{1}{2}M, \frac{1}{\sqrt{2}}KR + 2KR^2, \frac{1}{2}KM + KM^2\}$. $\qquad\qquad\square$

## C  Proof of Theorem 3.6

*Proof.* By Stein's identity $\mathbb{E}_{\boldsymbol{x} \sim p}[\mathcal{P}_{\boldsymbol{x}}\phi(\boldsymbol{x}, \boldsymbol{w})] = 0$, we have $\mathbb{E}_p f = 0$ for $\forall f \in \mathcal{F}_\infty$. This is because, assuming $f(\boldsymbol{x}) = \mathbb{E}_{\boldsymbol{w} \sim p_{\boldsymbol{w}}}[v(\boldsymbol{w})^\top \mathcal{P}_{\boldsymbol{x}}\phi(\boldsymbol{x}, \boldsymbol{w})]$,

$$\mathbb{E}_p f = \mathbb{E}_{\boldsymbol{x} \sim p}\mathbb{E}_{\boldsymbol{w} \sim p_{\boldsymbol{w}}}[v(\boldsymbol{w})^\top \mathcal{P}_{\boldsymbol{x}}\phi(\boldsymbol{x}, \boldsymbol{w})] = \mathbb{E}_{p_{\boldsymbol{w}}}[v(\boldsymbol{w})^\top \mathbb{E}_{\boldsymbol{x} \sim p}[\mathcal{P}_{\boldsymbol{x}}\phi(\boldsymbol{x}, \boldsymbol{w})]] = 0.$$

Therefore,

$$\mathbb{E}_{\hat{\mu}_{X^*}} f - \mathbb{E}_p f = \mathbb{E}_{\boldsymbol{x} \sim \hat{\mu}_{X^*}}[f(\boldsymbol{x})] = \mathbb{E}_{\boldsymbol{w} \sim p_{\boldsymbol{w}}}[\mathbb{E}_{\boldsymbol{x} \sim \hat{\mu}_{X^*}}[v(\boldsymbol{w})^\top \mathcal{P}_{\boldsymbol{x}}\phi(\boldsymbol{x}, \boldsymbol{w})]].$$

This gives

$$|\mathbb{E}_{\hat{\mu}_{X^*}} f - \mathbb{E}_p f| = \left| \sum_{j=1}^{d} \mathbb{E}_{\boldsymbol{w} \sim p_{\boldsymbol{w}}} [\mathbb{E}_{\boldsymbol{x} \sim \hat{\mu}_{X^*}} [v_j(\boldsymbol{w}) \mathcal{P}_{x^j} \phi(\boldsymbol{x}, \boldsymbol{w})]] \right|$$

$$\leq \sum_{j=1}^{d} |\mathbb{E}_{\boldsymbol{w} \sim p_{\boldsymbol{w}}} [\mathbb{E}_{\boldsymbol{x} \sim \hat{\mu}_{X^*}} [v_j(\boldsymbol{w}) \mathcal{P}_{x^j} \phi(\boldsymbol{x}, \boldsymbol{w})]]|$$

$$= \sum_{j=1}^{d} |\mathbb{E}_{\boldsymbol{w} \sim p_{\boldsymbol{w}}} [v_j(\boldsymbol{w}) g_{X^*, j}(\boldsymbol{w})]|.$$

Let $h_{X,j}(\boldsymbol{w}) = v_j(\boldsymbol{w}) g_{X,j}(\boldsymbol{w})$. Then Assumption 3.4.1-2 gives $\sup_{\boldsymbol{w}} |h_{X^*,j}(\boldsymbol{w}_\ell)| \leq \frac{\epsilon_j M_j}{\sqrt{m}}$, $\forall j = 1, \ldots, m$. We have

$$|\mathbb{E}_{\boldsymbol{w} \sim p_{\boldsymbol{w}}} [h_{X^*,j}(\boldsymbol{w})]| \leq |\mathbb{E}_{\boldsymbol{w} \sim p_{\boldsymbol{w}}} [h_{X^*,j}(\boldsymbol{w})] - \frac{1}{m} \sum_{\ell=1}^{m} h_{X^*,j}(\boldsymbol{w}_\ell)| + |\frac{1}{m} \sum_{\ell=1}^{m} h_{X^*}(\boldsymbol{w}_\ell)|$$

$$\leq \sup_{h_{X,j} \in v_j \mathcal{G}_j} |\mathbb{E}_{\boldsymbol{w} \sim p_{\boldsymbol{w}}} [h_{X,j}(\boldsymbol{w})] - \frac{1}{m} \sum_{\ell=1}^{m} h_{X,j}(\boldsymbol{w}_\ell)| + \frac{\epsilon_j M_j}{\sqrt{m}},$$

where $v_j \mathcal{G}_j = \{ \boldsymbol{w} \mapsto v_j(\boldsymbol{w}) g_{X,j}(\boldsymbol{w}) \colon X \in \mathcal{X}^n \}$. Therefore, we just need bound

$$\Delta_\ell(\boldsymbol{w}_1, \ldots \boldsymbol{w}_m) \stackrel{def}{=} \sup_{h_{X,j} \in v_j \mathcal{G}_j} |\mathbb{E}_{\boldsymbol{w} \sim p_{\boldsymbol{w}}} [h_{X,j}(\boldsymbol{w})] - \frac{1}{m} \sum_{\ell=1}^{m} h_{X,j}(\boldsymbol{w}_\ell)|.$$

This can be done using standard techniques in uniform concentration bounds. To do this, note that any $\boldsymbol{w}_\ell$ and $\boldsymbol{w}'_\ell$,

$$|\Delta_\ell(\boldsymbol{w}_1, \ldots, \boldsymbol{w}_\ell, \ldots, \boldsymbol{w}_m) - \Delta_\ell(\boldsymbol{w}_1, \ldots, \boldsymbol{w}'_\ell, \ldots, \boldsymbol{w}_m)|$$

$$\leq \frac{2 \sup_{h_{X,j} \in v_j \mathcal{G}_j} \sup_{\boldsymbol{w}} |h_{X,j}(\boldsymbol{w})|}{m} \leq \frac{2 V_j M_j}{m}$$

where we assume $\sup_{\boldsymbol{w}} |v_j(\boldsymbol{w})| = V_j$. By Mcdiarmid's inequality, we have

$$\Pr(\Delta_\ell(\boldsymbol{w}_1, \ldots, \boldsymbol{w}_m) > \mathbb{E}[\Delta_\ell(\boldsymbol{w}_1, \ldots, \boldsymbol{w}_m)] + t) \leq \exp(-\frac{m t^2}{2 V_j^2 M_j^2}).$$

On the other hand, the expectation $\mathbb{E}[\Delta_\ell(\boldsymbol{w}_1, \ldots, \boldsymbol{w}_m)]$ can be bounded by Rademacher complexity of $v_j \mathcal{G}_j$:

$$\mathbb{E}[\Delta_\ell(\boldsymbol{w}_1, \ldots, \boldsymbol{w}_m)] \leq 2 \mathcal{R}_m(v_j \mathcal{G}_j).$$

Restating the result, we have with probability $1 - \delta$, for $\forall \delta > 0$,

$$\sup_{h_{X,j} \in v_j \mathcal{G}_j} \left| \mathbb{E}_{\boldsymbol{w} \sim p_{\boldsymbol{w}}} [h_{X,j}(\boldsymbol{w})] - \frac{1}{m} \sum_{\ell=1}^{m} h_{X,j}(\boldsymbol{w}_\ell) \right| \leq 2 \mathcal{R}_m(v_j \mathcal{G}_j) + V_j M_j \sqrt{\frac{2 \log(1/\delta)}{m}}.$$

Overall, this gives

$$|\mathbb{E}_{\hat{\mu}_{X^*}} f - \mathbb{E}_p f| = \sum_{j=1}^{d} |\mathbb{E}_{\boldsymbol{w} \sim p_{\boldsymbol{w}}} [h_{X^*,j}(\boldsymbol{w})]|$$

$$\leq \sum_{j=1}^{d} \left( 2 \mathcal{R}_m(v_j \mathcal{G}_j) + V_j M_j \sqrt{\frac{2 \log(1/\delta)}{m}} + \frac{\epsilon_j M_j}{\sqrt{m}} \right).$$

$$= \frac{1}{\sqrt{m}} \left( V M \sqrt{2 \log(1/\delta)} + \epsilon M \right) + 2 \sum_{j=1}^{d} \mathcal{R}_m(v_j \mathcal{G}_j),$$

where we use the fact that $V^2 = \sum_{j=1}^{d} V_j^2$, $M^2 = \sum_{j=1}^{d} M_j^2$ and $\epsilon^2 = \sum_{j=1}^{d} \epsilon_j^2$.

We just need to bound the Rademacher complexity $\mathcal{R}_m(v_j\mathcal{G}_j)$. This requires recalling some properties of Rademacher complexity. Let $\{\mathcal{F}_j \colon j = 1, \ldots, n\}$ be a set of function sets, and $\frac{1}{n}\sum_{j=1}^n \mathcal{F}_i$ be the set of functions consisting of functions of form $\frac{1}{n}\sum_{j=1}^n f_i, \forall f_i \in \mathcal{F}_i$. Then we have (see, e.g., Bartlett & Mendelson (2002))

$$\mathcal{R}_m\Big(\frac{1}{n}\sum_{j=1}^n \mathcal{F}_i\Big) \le \frac{1}{n}\sum_{i=1}^n \mathcal{R}_m(\mathcal{F}_i).$$

Applying this to $\mathcal{G}_j$, we have

$$\mathcal{R}_m(\mathcal{G}_j) \le \mathcal{R}_m(\mathcal{P}_j\Phi) \le \frac{R_j}{\sqrt{m}}.$$

Further, applying Lemma C.1.5) below, we have

$$\mathcal{R}_m(v_j\mathcal{G}_j) \le 2(\mathcal{R}_m(\mathcal{G}_j) + \frac{V_j}{\sqrt{m}})(M_j + V_j) \le \frac{2}{\sqrt{m}}(R_j + V_j)(M_j + V_j) \le \frac{2}{\sqrt{m}}(R_j^2 + 2V_j{}^2 + M_j^2)$$

Therefore,

$$\sum_{j=1}^d \mathcal{R}_m(v_j\mathcal{G}_j) \le \frac{2}{\sqrt{m}}(R^2 + 2V^2 + M^2).$$

Putting everything together, we get

$$|\mathbb{E}_{\hat{\mu}_{X^*}}f - \mathbb{E}_p f| \le \frac{1}{\sqrt{m}}\left(VM\sqrt{2\log(1/\delta)} + \epsilon M + 2R^2 + 4V^2 + 2M^2\right).$$

This concludes the proof. $\qquad\qquad\square$

## C.1 Rademacher Complexity

The following Lemma collects some basic properties of Rademacher complexity. See Bartlett & Mendelson (2002) for more information.

For a function set $\mathcal{F}$, its Rademacher complexity is defined as

$$\mathcal{R}_m(\mathcal{F}) = \mathbb{E}\left[\sup_{f\in\mathcal{F}}\left|\frac{1}{m}\sum_{i=1}^m \sigma_i f(x_i)\right|\right],$$

where the expectation is taken when $\sigma_i$ are i.i.d. uniform $\{\pm 1\}$-valued random variables and $x_i$ are i.i.d. random variables from some underlying distribution. A basic property of Rademacher complexity is that

$$\mathbb{E}\left[\sup_{f\in\mathcal{F}}\left|\frac{1}{m}\sum_{i=1}^m f(x_i) - \mathbb{E}f\right|\right] \le 2\mathcal{R}_m(\mathcal{F}).$$

*Proof.*

$$\mathbb{E}\left[\sup_{f\in\mathcal{F}}\left|\frac{1}{m}\sum_{\ell=1}^m f(x_i) - \mathbb{E}f\right|\right] \le \mathbb{E}\left[\sup_{f\in\mathcal{F}}\left|\frac{1}{m}\sum_{i=1}^m (f(x_i) - f(x_i'))\right|\right]$$

$$= \mathbb{E}\left[\sup_{f\in\mathcal{F}}\left|\frac{1}{m}\sum_{i=1}^m \sigma_i(f(x_i) - f(x_i'))\right|\right]$$

$$\le 2\mathbb{E}\left[\sup_{f\in\mathcal{F}}\left|\frac{1}{m}\sum_{i=1}^m \sigma_i f(x_i)\right|\right]$$

$$= 2\mathcal{R}_m[\mathcal{F}].$$

$\square$

**Lemma C.1.** *Let $\mathcal{F}$, $\mathcal{F}_1$ and $\mathcal{F}_2$ are real-valued function classes.*

*1) Define $\mathcal{F}_1 + \mathcal{F}_2 = \{f + g\colon f \in \mathcal{F}_1,\ g \in \mathcal{F}_2\}$. We have*

$$\mathcal{R}_m(\mathcal{F}_1 + \mathcal{F}_2) \leq \mathcal{R}_m(\mathcal{F}_1) + \mathcal{R}_m(\mathcal{F}_2).$$

*2) Let $\phi\colon \mathbb{R} \to \mathbb{R}$ be an $L_\phi$-Lipschitz function. Define $\phi \circ \mathcal{F} = \{\phi \circ f\colon \forall f \in \mathcal{F}\}$. We have*

$$\mathcal{R}_m(\phi \circ \mathcal{F}) \leq 2L_\phi \mathcal{R}_m(\mathcal{F}) + \frac{\phi(0)}{m}.$$

*3) For any uniformly bounded function $g$, we have*

$$\mathcal{R}_m(\mathcal{F} + g) \leq \mathcal{R}_m(\mathcal{F}) + \frac{||g||_\infty}{\sqrt{m}}.$$

*4) For constant $c \in \mathbb{R}$ and $c\mathcal{F} = \{x \mapsto cf(x)\colon \forall f \in \mathcal{F}\}$,*

$$\mathcal{R}_m(c\mathcal{F}) = |c|\mathcal{R}_m(\mathcal{F}).$$

*5) Define $g\mathcal{F} = \{x \mapsto f(x)g(x)\colon \forall f \in \mathcal{F}\}$. Assume $||\mathcal{F}||_\infty := \sup_{f \in \mathcal{F}} ||f||_\infty < \infty$, we have*

$$\mathcal{R}_m(g\mathcal{F}) \leq 2(\mathcal{R}_m[\mathcal{F}] + \frac{||g||_\infty}{\sqrt{m}})(||\mathcal{F}||_\infty + ||g||_\infty).$$

*Proof.* 1) - 4) are standard results; see Theorem 12 in Bartlett & Mendelson (2002).

For 5), note that

$$fg = \frac{1}{4}(f + g)^2 - \frac{1}{4}(f - g)^2.$$

3) gives

$$\mathcal{R}_m(\mathcal{F} \pm g) \leq \mathcal{R}_m[\mathcal{F}] + \frac{||g||_\infty}{\sqrt{m}}$$

Further, note that $\phi(x) = x^2$ is $2(||F||_\infty + ||g||_\infty)$-Lipschitz on interval $[-||F||_\infty - ||g||_\infty, ||F||_\infty + ||g||_\infty]$. Applying 2) and then 1) and 4) gives

$$\mathcal{R}_m(g\mathcal{F}) \leq 2(||\mathcal{F}||_\infty + ||g||_\infty)(\mathcal{R}_m(\mathcal{F}) + \frac{||g||_\infty}{\sqrt{m}}).$$

$\square$

Our results require bounding the Rademacher complexity $\mathcal{R}_m(\mathcal{P}_j\Phi)$ of the Steinalized features, $\mathcal{P}_j\Phi = \{w \mapsto \mathcal{P}_{x^j}\phi(\boldsymbol{x}, w)\colon \boldsymbol{x} \in \mathcal{X}\}$. The following result bounds the Rademacher complexity of the Steinalized set using the complexity of the original feature set and its gradient set.

**Lemma C.2.** *Define $\Phi = \{\boldsymbol{w} \mapsto \phi(\boldsymbol{x}, \boldsymbol{w})\colon \forall \boldsymbol{x} \in \mathcal{X}\}$ and $\nabla_j\Phi = \{\boldsymbol{w} \mapsto \nabla_{x^j}\phi(\boldsymbol{x}, \boldsymbol{w})\colon \forall \boldsymbol{x} \in \mathcal{X}\}$. Then*

$$\mathcal{R}_m(\mathcal{P}_j\Phi) \leq ||\nabla_{\boldsymbol{x}_\ell} \log p||_\infty \mathcal{R}_m(\Phi) + \mathcal{R}_m(\nabla_j\Phi),$$

*where $||\nabla_{\boldsymbol{x}_\ell} \log p||_\infty = \sup_{\boldsymbol{x} \in \mathcal{X}} |\nabla_{\boldsymbol{x}_\ell} \log p(\boldsymbol{x})|$.*

## D   Empirical Experiments

Our results show that linear features allow us to obtain accurate estimates of the first and second moments for Gaussian-like distributions, while random features can obtain a good overall distributional approximation with high probability. To test these theoretical observations empirically, we design a "linear+random" kernel:

$$k(\boldsymbol{x}, \boldsymbol{x}') = \alpha(1 + \boldsymbol{x}^\top \boldsymbol{x}') + \beta \sum_{\ell = d+2}^{n} \phi(\boldsymbol{x}, \boldsymbol{w}_\ell)\phi(\boldsymbol{x}', \boldsymbol{w}_\ell),$$

Figure 1: Results on standard Gaussian distribution ($d = 100$). (a)-(b) show the MSE when using the obtained particles to estimate the mean and second order moments of each dimension, averaged across the dimensions. (c) shows the maximum mean discrepancy between the particle distribution and true distribution. (d) shows the average values of the estimated variance (the true variance is 1).

Figure 2: (a)-(b) Results on random 100 dimensional non-spherical Gaussian distributions whose covariance matrix has a conditional number of $\lambda_{\max}/\lambda_{\min} = 10$. (c)-(d) The performance on random non-spherical Gaussian distributions with different conditional numbers. Results averaged on 20 random models.

where we take $\alpha = 1/(d+1)$ and $\beta = 1/(n-d-1)$ in our experiments. In the case when there are fewer particles than dimension plus one ($n \leq d+1$), we have $k(\boldsymbol{x}, \boldsymbol{x}') = 1 + \boldsymbol{x}^\top \boldsymbol{x}'$, which only include the linear features, and when $n > (d+1)$, additional random features are added, so that the total number of features matches the number of particles.

We take $\phi(\boldsymbol{x}, \boldsymbol{w})$ to be the random cosine feature in (6) to approximate the Gaussian RBF kernel. Note that in our method, the random parameters $\{\boldsymbol{w}_\ell\}$ are drawn in the beginning and fixed across the iterations of the algorithm, but we adopt the bandwidth $h$ across the iterations using the median trick. We compare exact Monte Carlo with SVGD with different kernels, including the standard Gaussian RBF kernel, the linear kernel $k(\boldsymbol{x}, \boldsymbol{x}') = 1 + \boldsymbol{x}^\top \boldsymbol{x}'$, and the linear+random kernel defined above.

**Gaussian Models** We start with verifying our theory on a simple standard Gaussian distribution $p(\boldsymbol{x}) = \mathcal{N}(\boldsymbol{x}, 0, I)$ with $d = 100$ dimensions. In Figure 1, we can see that all SVGD methods estimate the mean parameters exceptionally well (Figure 1(a)). Variance estimation is more difficult for SVGD in general, but both the Linear+Random and Linear kernels perform well as the theory predicts: the errors drop quickly as $n$ approaches $d+1$ (the minimum particle size needed to recover mean and covariance matrices), and only the numerical error is left when $n > d+1$.

To examine the variance estimation more closely, we show in Figure 1(d) the value of the estimated variance (averaged across the dimensions) on the same 100-dimensional standard Gaussian distribution. We find that all the variants of SVGD tend to underestimate the variance when there is insufficient number of particles (in particular, when $n < d+1$), but the kernels that include linear features give (near) exact estimation once $n \geq d+1$.

Figure 2 shows a similar plot for 100-dimensional non-spherical Gaussian distributions when the conditional number of the covariance matrix varies. In particular, we set $p(\boldsymbol{x}) = \mathcal{N}(\boldsymbol{x}; \boldsymbol{\mu}, \Sigma)$ where $\boldsymbol{\mu} \sim \mathrm{Unif}([-3, 3])$ and $\Sigma = I + \alpha \Lambda \Lambda^\top$, with the elements of $\Lambda$ drawn from $\mathcal{N}(0, 1)$ and $\alpha$ adjusted to make the conditional number $\lambda_{\max}/\lambda_{\min}$ of $\Sigma$ equal specific numbers. When the condition number equals 1, we should have $\Sigma = I$.

Figure 2(a)-(b) show the estimation of the first and second order moments when the conditional number equals 10, in which SVGD(linear+random) and SVGD(linear) again show a near exact

recovery after $n > d + 1$. Figure 2(c)-(d) show that as the conditional number increases, the accuracy of all the methods decreases, but SVGD(linear+random) and SVGD(linear) still significantly outperform Monte Carlo estimation. The increased errors in SVGD(linear+random) and SVGD(linear) are caused by the increase of numerical error because it is more difficult to satisfy the fixed point equation with high accuracy when the conditional number is large.

Figure 3: Results on Gaussian mixture models $p(\boldsymbol{x}) = \sum_{k=1}^{15} \mathcal{N}(\alpha\boldsymbol{\mu}_k, I)$, where $\boldsymbol{\mu}_k \sim$ Uniform$([0, 1])$ and $\alpha$ controls the Gaussianity of $p$ (when $\alpha = 0$, $p$ is standard Gaussian). All the results are the relative performance w.r.t. exact Monte Carlo sampling method with the sample size (we fix for all the methods). We fix $n = 100$ for all the methods and average the result over 20 random models.

Figure 4: Results on randomly generated Gaussian-Bernoulli RBM, averaged on 20 trials.

**Gaussian Mixture Models**  We consider a Gaussian mixture model with density fucntion $p(\boldsymbol{x}) = \frac{1}{15} \sum_{j=1}^{15} \mathcal{N}(\boldsymbol{x}; \alpha\boldsymbol{\mu}_j, I)$, where $\boldsymbol{\mu}_j$ is randomly drawn from Unifrom$([0, 1])$, and $\alpha$ can be viewed as controlling the Gaussianity of $p(\boldsymbol{x})$: when $\alpha$ equals zero, $p(\boldsymbol{x})$ reduces to the standard Gaussian distribution, while when $\alpha$ is large, $p(\boldsymbol{x})$ would be highly multimodal with mixture components far away from each other.

Figure D shows the relative performance of SVGD with different kernels compared to exact Monte Carlo sampling. We find that SVGD methods generally outperform Monte Carlo unless $\alpha$ is very large. In Figure D(b), we can see that SVGD(Linear) outperforms SVGD(RBF) when $p$ is close to Gaussian (small $\alpha$), and performs worse than SVGD(RBF) when $p$ is highly non-Gaussian (large $\alpha$). SVGD(Linear+Random) combines the advantages of both and tends to match the best of SVGD(Linear) and SVGD(RBF) in all the range of $\alpha$.

**Gaussian-Bernoulli RBM**  Gaussian-Bernoulli RBM is a hidden variable model consisting of a continuous observable variable $\boldsymbol{x} \in \mathbb{R}^d$ and a binary hidden variable $\boldsymbol{h} \in \{\pm 1\}^{d'}$ with probability

$$p(\boldsymbol{x}, \boldsymbol{h}) \propto \sum_{\boldsymbol{h} \in \{\pm\}^{d'}} \exp(\boldsymbol{x}^\top B \boldsymbol{h} + \boldsymbol{b}^\top \boldsymbol{x} + \boldsymbol{c}^\top \boldsymbol{h} - \frac{1}{2}||\boldsymbol{x}||_2^2),$$

where we randomly draw $\boldsymbol{b}$ and $\boldsymbol{c}$ from $\mathcal{N}(0, I)$, and the elements of $B$ from Uniform$(\{\pm 0.1\})$. We use $d = 100$ observable variables and $d' = 10$ hidden variables, so $p(\boldsymbol{x})$ is effectively a Gaussian mixture with $2^{10}$ components. The results are shown in Figure D, where we find that SVGD(Linear+Random) again achieves the best performance in terms of all the evaluation metrics.