[Reviews · NeurIPS 2018]

Reviewer 1



The work analyzes fixed-point solutions of the Stein Variational Gradient Descent (SVGD) objective for finite number of particles. It shows that SVGD exactly estimates expectations for a set of functions, the "Stein matching set", determined by the kernel. For linear kernels applied to Gaussian distributions, this set includes all second-order polynomials; thus linear-kernel SVGD estimates the mean and variance of Gaussians exactly. More generally, random features allow probabilistic bounds on distribution approximation. The results appear both theoretically interesting and of potential practical value in designing inference algorithms. The questions asked are natural (I remember wondering myself about connections to sigma points when reading one the first SVGD papers), and the answers are neat and slightly counterintuitive -- one might have expected that an RBF kernel rather than a linear kernel would estimate Gaussian moments well. The writing is generally clear, though there are minor typos and grammatical errors that might benefit from proofreading by a native English speaker. I found no obvious technical mistakes, though I did not have time to verify the math in any depth. I appreciate that the authors include basic experimental verification in the supplemental. Overall, this is a solid theory paper that I think many NIPS attendees will find interesting. nits (incomplete): line 6: 'image of Stein operator' -> 'image of the Stein operator' line 21: 'losses' -> 'loses'

Reviewer 2



In "Stein Variational Gradient Descent as Moment Matching," the authors first introduce the algorithm known as Stein Variational Gradient Descent (SVGD). While some work has been done trying to provide a theoretical analysis of this method, the consistency of SVGD is largely still open for finite sizes of n. (In fact, the analysis is still completely open when the domain is not compact either.) The authors of this paper make headway on this problem. By studying the fixed point solution to SVGD, they show there are a set of functions for which the the fixed point solution perfectly estimates their mean under the target distribution (they call this the Stein set of functions). They argue that using a polynomial kernel when the target is a Gaussian will force any fixed point solution of SVGD to exactly estimate the mean and covariance of the target distribution, assuming the SVGD solution points are full rank. The major contribution of this paper is that by studying the properties of finite dimensional kernels, they are able to employ random Fourier features to provide a theoretical analysis of the fixed points for these "randomized" kernels. By using this formulation of a randomized kernel Stein discrepancy, the authors provide a VC-like bound on the (randomized) kernel Stein discrepancy's value at a fixed point solution that converges at a O(m^{-1/2}log^{3/2}m) rate, where m indexes the number of random Fourier features. Their results do also require the domain to be compact, but the work does appeal to the finite sample regime and thus is a solid theoretical finding. My biggest critique of this paper are the assumptions in 3.4. Because they have modified the kernel with a randomized, low rank approximation, their results do not immediately carry over to the non-randomized version. For example, for the Gaussian kernel and Gaussian target on a truncated or unbounded domain, can the assumptions of 3.4 be checked? I would be very interested in knowing what conditions on n (the number of SVGD samples), m, the target distribution P, and the kernel k are enough to assure the first assumption of 3.4 holds. While I would love to see this included in the paper, I think the rest of the paper is strong enough to merit publication at the conference. ** Quality L30-31, 110-111: Perhaps it is worth emphasizing here that these results typically only hold on compact domains as well? Eqn 2, L103, Eqn 9, etc.: Isn't k the kernel for H_0 and thus how can the Stein operator be defined for a R-valued function? Eqn 8: Where is P_{x_j} defined? I think it might be more clear to leave P_x for this. L158-171: These lines seem a bit too generic to be included in this paper in my opinion. There are formulations which are solutions to Stein's equation, e.g., see Theorem 5 of Gorham et al "Measuring Sample Quality with Diffusions." Studying solutions to Stein's equation is an active field of research. L224-225: I can't tell how strong this assumption is. Most importantly, are fixed point solutions for SVGD in the original kernel "close" to the fixed point solutions to SVGD in the random Fourier kernel? That piece seems necessary to push the theory forward for a non-randomized kernel. ** Clarity The paper is for the most part clearly written. There is some sloppiness where the authors commonly apply the Stein operator to a R-valued function, but otherwise the ideas are well explained. I recommend the authors address this issue throughout the paper. ** Originality This paper does identify some new ways to view the kernel Stein discrepancy and uses those to provide a finite sample analysis of a related, randomized problem. The theory here does only apply to these "randomized" kernels (via the random Fourier formulation), while the kernels used in practice are not random. Hence there is a little left to be shown, in particular, that the fixed point solutions are not too different between the randomized kernel and the original (nonrandom) kernel. See the last paragraph of the summary section. ** Signifance SVGD has become a popular (though expensive) black box method for doing posterior inference, despite it having no great theoretical guarantees. This paper provides the best analysis of this method to date in the finite sample regime and thus is a nice contribution.

Reviewer 3



In this paper, an analysis of Stein variational gradient descent (SVGD) as a moment matching was given. I did not go through the proofs attached in the supplementary. From my experience, SVGD seems to be a promising framework for the synthesis of MCMC and VI, yet there is room for further analysis and development. This work seems to fill some gap by providing description of the 'fixed point' characteristics of SVGD. Specifically, I think the contribution of the paper is significant since it managed to provide analysis when the number of samples does not asymptotically grow to infinity. Such analysis makes more sense because SVGD does not add samples as time grow, contrary to the case of MCMC. Editorial comments: - The variable \epsilon is used twice for indicating the 'step size' of SVGD and size of neighborhood in Lemma 3.1. One of them should be changed for consistency of writing. - \mathtt{Stein Equation} in equation (13) seems unnecessary. - Readability of equation between line 214 and 215 can be improved, e.g., use consistent spacing and make the word 's.t.' into non-italic. Typos: - replace 'losses' by 'looses' in line 21. - insert space in 'upto' in line 180. - erase square bracket in equation above line 218.

Reviewer 4



In this paper, the authors study the fixed point properties of Stein Variational Gradient Descent, an nonparametric inference algorithm which updates the positions of a set of particles in the direction of steepest descent of the KL functional, subject to a max norm in a RKHS. Until this paper, properties of SVGD were understood in the limit of a large number of particles; here, the authors study instead the properties of any fixed point of the SVGD updates with a fixed number of particles. They identify that fixed points define an atomic distribution which matches the moment of the target distribution on a set of function they call the Stein Matching Set - more precisely, the images of the elements of the RKHS (evaluated at the atoms of the fixed point) through the Stein Operator. That definition being implicit (understanding the set of functions whose moments are matched requires knowing the values of the atoms of the fixed point), they characterize more precisely for degenerate kernels. They show that for linear features and Gaussian kernels, this correspond to matching the first two moments. Finally, they study kernels defined through random features and find that the fixed points have an error in sqrt(1/n) in kernelized stein discrepancy. This paper is a very nice read; it is self-contained, excellently written, and provides interesting insights on the behavior of SVGD. While most of the results (barring section 3.4, more complex) are perhaps not entirely surprising, they are novel to me and shed light both on what we can expect of SVGD inference schemes, and more importantly, how to design the features used so that good approximations are obtained on the desired test functions (particularly in the case of degenerate kernels, though they are less powerful approximators). Minor criticism: section 3.4 seems like it contains important results, but is fairly condensed. It makes it both harder to read and, as a result of not being 'contextualized' enough, harder to appreciate: how do theorem 3.5 and 3.6 compare to similar frameworks in statistical learning theory? What was the state of knowledge regarding uniform bounds using these types of features? Was the 1/sqrt(n) rate achieved by other methods, can it be improved upon, etc.? Maybe tightening the section by clearly presenting the main result and its broader significance, and pushing the rest of the section in the appendix would .have been preferable, leaving some room for simple numerical experiments (only found in the appendix so far). Nevertheless, I think this paper is a valuable addition to the body of work on SVGD, with clear presentation and intuitive results.